# The Enhanced and Tunable Sustained Release of Pesticides Using Activated Carbon as a Carrier

**DOI:** 10.3390/ma12234019

**Published:** 2019-12-03

**Authors:** Jun Yang, Wanyu Zang, Zheng Zhang, Peng Wang, Qing Yang

**Affiliations:** 1School of Bioengineering, Dalian University of Technology, Dalian 116024, China; zangwanyu@mail.dlut.edu.cn (W.Z.); alexamde@mail.dlut.edu.cn (Z.Z.); 408994882@mail.dlut.edu.cn (P.W.); 2Institute of Plant Protection, Chinese Academy of Agricultural Sciences, Beijing 100081, China

**Keywords:** activated carbon, sustained release, dopamine, nitrogen-doped activated carbon, pest control

## Abstract

The sustained release of pesticides improves drug utilization efficiency and reduces their adverse effects. Activated carbon (AC) is an excellent adsorbent and promising soil conditioner. It has a rich, porous structure and thus can store and gradually release drugs. In this study, three AC materials with surface areas ranging from 800–2000 m^2^/g were used and two types of modified activated carbons were prepared, and their capacity as drug carriers was evaluated by using 2,4-Dichlorophenoxyacetic acid sodium (2,4-D sodium) as the model pesticide. The preparations were characterized by scanning electron microscopy, nitrogen physical analysis, and zeta potential. The five preparations showed an enhanced and tunable sustained release of drugs. AC1, with the highest specific surface area, possesses the best drug-loading capacity, reaching 679.18 mg/g, but the lowest drug release rate of 32.31% in 96 h. PDA-AC3 has the lowest specific surface area, showing limited drug-loading ability, 82.94 mg/g, but 100% drug release within 72 h. This study suggests that activated carbon has potent applications in agricultural pest control as an inexpensive, effective, controllable, and safe pesticide carrier.

## 1. Introduction

Pesticides are widely used to remove weeds and to control pests and plant diseases. However, depending on the application and environmental conditions, more than 90% of applied pesticides are lost in the environment or fail to reach their target areas, which not only increases the cost but may also adversely affect the environment and human health [1]. The long-term use of herbicides in agricultural production may lead to the presence of pesticide residues and other harmful substances in water, soil, and food. The toxic ingredients accumulated in a certain part of the human body will also cause great damage to human organs.

Improvements in the effectiveness of pesticides and minimization of pollution has always been an important issue in the field of agriculture and environmental chemistry. Pesticide formulations optimized for controlled release provide a practical solution. Substances such as starch [2], chitosan [3,4], clay [5], lignin [6], sodium alginate [7], and synthetic polymers [8,9,10,11] have been studied as carriers for preparation of pesticide sustained release formulations. However, these preparations have problems of high cost, poor adsorption performance, excessive sustained release rate, and/or recalcitrance to degradation.

Activated carbon (AC) is a form of carbon with a turbostratic structure, with rich surface functional groups and developed micropores [12,13,14,15,16,17] and a high specific surface area of 500–3000 m^2^/g [18,19]. AC has been used as a pharmaceutical carrier and has the advantage of being inexpensive, commercially available, non-toxic, and highly absorbable [20]. AC is also a promising soil conditioner and is commonly used to enhance soil water and gas permeability and improve soil pH and redox conditions and is thus a potentially environmentally friendly pesticide carrier material [21,22,23,24,25,26].

Activated carbon possesses perfect adsorption ability and has been widely used to absorb a variety of chemicals [27,28,29,30,31,32,33]. However, the desorption of chemicals from activated carbon has always been a challenge, due to the high affinity of the compounds to the sorbent surface [29,34]. Additional physical and chemical treatments are normally required, such as heating, ultrasound or using NaOH solution to release chemicals from the sorbent [34]. Obviously, these methods are not adapted to agricultural applications. Moreover, the complete release of pesticides from carriers during a plant growth season also matters because it may reduce the risk of pesticide residue in soil, which may otherwise cause drug resistance or damage in subsequent crops.

Dopamine is a common catechol-containing neurotransmitter that can be oxidized to hydrazine and then produce a dark brown-black insoluble biopolymer, polydopamine (PDA), which is available for immobilization at the organic, inorganic, and superhydrophobic surface to form a thin layer. In this study, a commonly used herbicide, 2,4-Dichlorophenoxyacetic acid (2,4-D) sodium, was used as the model chemical. 2,4-D sodium has the advantages of low cost, wide application, and low toxicity [35]. However, because 2,4-D sodium is highly water-soluble, it is easily washed away by rain, and the volatiles are transferred to the environment [36], which not only harms the health of non-target plants and animals, but also harms humans [37,38,39,40]. Three AC and two types of PDA-modified AC materials were chosen as the potential sustained-release carriers, and their capability for adsorption and sustained release of 2,4-D sodium was investigated. The chemical groups, such as amines from PDA, are introduced after modification, improving the interaction between drugs and AC material. Our study suggests that activated carbon has potent applications in agricultural pest control as an inexpensive, effective, and safe pesticide carrier.

## 2. Material and Methods

### 2.1. Materials

Three ACs with different specific surface areas were selected: AC1, AC2, and AC3, with specific surface areas of >2000, 1000–2000, and <1000 m^2^ g^−1^, respectively. AC1 and AC2 were kindly provided by Professor Chang Yu, Carbon Research Laboratory, Dalian University of Technology [41]. The AC3 preparations used in this study were purchased from Sigma (Saint Louis, MO, USA). 2,4-D sodium and Dopamine hydrochloride were purchased from Aladdin (98%, Shanghai, China). 3-hydroxymethyl aminomethane was purchased from Solarbio (Beijing, China). Deionized water was obtained from a Milli-Q water system (Master Q15, Hitech Instruments CO., Shanghai, China) and was used for all reactions and treatment processes.

### 2.2. Loading of 2,4-D Sodium into the AC

After preparation of a 10.0 mg/mL 2,4-D sodium solution, 20 mL 2,4-D sodium solution and 0.10 g carrier material were added to several 50-mL centrifuge tubes, mixed and placed on a constant temperature shaker. These were vortexed for 24 h at a constant rate of 200 rpm and a constant temperature of 25 ± 2 °C. A syringe was used to aspirate a 300 μL sample through a 0.22 μm microporous membrane [42], and the concentration of 2,4-D sodium was measured by a UV spectrometer Infinite 200 Pro (Tecan Austria GmbH, Untersbergst, Austria) at 282 nm [43]. The corresponding 2,4-D sodium concentration, Co, was calculated from a standard curve, and three replicates were taken for parallel determination using the following equation:*Qe = (Co − Ce) V/M*,(1)
where *Qe* is the drug loading of the carrier material with respect to 2,4-D sodium at time t, mg/g; *Co* is the initial concentration of 2,4-D sodium, mg/mL; *Ce* is the concentration of 2,4-D sodium remaining at time *t*, mg/mL; *V* is the solution volume, mL; and *M* is the mass of the carrier material, g.

### 2.3. Modification of AC3 with Dopamine Hydrochloride

A 200 mL 3-hydroxymethyl aminomethane solution (2 mg/mL) was prepared. Then, 1.0 g activated carbon powder was added to the 3-hydroxymethyl aminomethane solution and dispersed uniformly by ultrasonic means for 30 min. Subsequently, 200 mg dopamine hydrochloride was added. The solution was stirred at room temperature for 24 h at 400 rpm to form the dopamine layer. Finally, the modified activated carbon sample PDA-AC3 was obtained by centrifugation at a speed of 17,000 rpm. This was washed three times with deionized water and dried.

### 2.4. Preparation of Nitrogen-Doped Activated Carbon NC3

PDA-AC3 can be further carbonized by firing and artificial pore-forming to produce nitrogen-doped activated carbon NC3. PDA-AC3 was placed in a tubular furnace filled with nitrogen to start the heating program. The temperature was raised to 800 °C at a heating rate of 5 °C/min and was held for 2 h. Activated carbon samples (NC3) were collected after carbonization.

### 2.5. Characterization

#### 2.5.1. Scanning Electron Microscope (SEM)

The surface morphology of the samples was observed using a field-emission scanning electron microscope (FEI NOVA NanoSEM 450, FEI, Portland, OR, USA). Before the test, the powder sample was fixed directly on the conductive carbon tape, and nitrogen was sprayed for 1 min. During the test, two different magnifications were selected from the same carrier sample for observation.

#### 2.5.2. Nitrogen Adsorption Measurements

Nitrogen adsorption isotherms were measured by a Micromeritics 3Flex 3500 instrument (Micromeritics, Norcross, GA, USA). Before the measurements, the samples were degassed under vacuum for 6 h at 200 °C. The specific surface area was calculated from adsorption data in the relative pressure range of 0.05–0.35 using the standard Brunauer–Emmett–Teller (BET) method. Pore size distributions for different activated carbons calculated using Density Functional Theory (DFT) software. The micropore volumes (V_mic_) and micropore areas (S_mic_) were calculated using the t-plot method; the total pore volumes (V_total_) were obtained from the last point of the isotherm at a relative pressure of 0.99; and the pore size distributions were determined using the density functional theory method [44,45].

#### 2.5.3. Zeta Potential

0.1 mg/mL AC sample suspensions were prepared, and the zeta potential was analyzed using a Malvern Nanoparticle Size Analyzer (ZEN3690, Malvern, UK).

### 2.6. Release of 2,4-D Sodium

The preparation was centrifuged at 6000 g for 15 min, after which the supernatant was poured off and dried at 40 °C for 48 h. A total of 0.10 g of drug-loaded ACs were weighed and dispersed into 20 mL of water, transferred to a dialysis bag, and placed in a beaker containing 300 mL deionized water (pH = 7). The preparation was released at room temperature, and sampling was timed to measure the 2,4-D sodium concentration in the beaker. The control group was configured with the same concentration of 2,4-D sodium solution as the experimental group, and the operation was the same. Three replicates were measured for parallel determination. For assay analysis, 1 mL of sample was added each time to 1 mL of water in a beaker, and the volume of the solution in the beaker was held constant at 300 mL:*Er* = *Ct* × *V*_0_/*Mp*,(2)
where *Er* is the cumulative release of 2,4-D sodium, %; *Ct* is the concentration of 2,4-D sodium in the release medium at time n, mg/mL; *V*_0_ is the volume of the released solution, mL; and *Mp* is the mass of 2,4-D sodium in the carrier, mg.

## 3. Results and Discussion

### 3.1. Structure of the Five ACs

The five ACs have amorphous lamellar structures, with irregular accumulation between the layers, and an internal structure with a large specific surface area (Figure 1, Table 1). The AC3 carbon has the most developed void structure and a large pore size; the pore collapse of modified PDA-AC3 is serious, and the specific surface area becomes the smallest. The specific surface area and pore volume of modified NC3 are close to those of AC3. The pore sizes of the five activated carbons are mainly distributed in the regions of 0.5–2 nm. N_2_ adsorption–desorption measurements are performed to study the specific surface area and pore size distribution of the AC. All five kinds of AC belonged to the Type I adsorption curve, and the low-pressure end of the Y-axis indicates that the material had a strong force with nitrogen (Figure 2). The more micropores present, the stronger the adsorption capacity is. Because AC1 had the largest specific surface area and the largest number of micropores, its adsorption capacity is the strongest of the studied ACs. PDA-AC3 has the lowest surface area, and thus its adsorption capacity is the worst (Table 1).

Moreover, NC3 carbon was prepared from carbonization of 3-hydroxymethyl aminomethane and dopamine adsorbed on AC3. The adsorption parameters clearly indicate the reduction in micropore and total pore volumes and surface area. This results from the deposition of carbon from the N-containing precursors within the pores of AC3, leading to partial pore blockage.

### 3.2. The Modification of AC3

Carbon, hydrogen, and nitrogen elemental analysis verified that AC3 has been successfully modified. As shown in Table 2, activated carbon AC3 mainly contains 88.15% carbon, and a small amount of nitrogen and hydrogen. The percentage of nitrogen in unmodified AC3 is 0.636%, the percentage of nitrogen in modified PDA-AC3 is 1.997%, and the percentage of nitrogen in NC3 is 1.694%. It can be seen that the content of nitrogen element increases obviously after modification. However, NC3 has less nitrogen, which may due to be the conversion of some nitrogen to carbon after high temperature carbonization or the loss of some unstable amino groups.

### 3.3. Adsorptive Property of 2,4-D Sodium into ACs

Figure 3 shows the adsorption capacity of five AC preparations for 2,4-D sodium in aqueous solution at 25 °C. AC1, AC2, and ACs exhibit good adsorption performance for the 2,4-D sodium, and the drug loading amount reaches 300 mg/g or more. The adsorption mode of AC is mainly physical adsorption, which is related to the specific surface area and pore volume of the material. The adsorption capacity of AC is greater, with a large specific surface area and pore volume. Because the specific surface area and pore volume of AC1 are the largest, the loading amount of AC1 is the greatest, reaching 679.18 mg/g.

After dopamine modification, the drug loading of PDA-AC3 is dramatically decreased to 82.94 mg/g. The decrease in drug loading of PDA-AC3 may be due to the blockage of pore channels caused by dopamine modification and the decrease of specific surface area and pore volume of activated carbon. It may also be because the modification of dopamine hydrochloride improves the hydrophilicity of activated carbon; thus, the adsorption sites on the surface of activated carbon are occupied by water molecules, which reduce the adsorption capacity of activated carbon for the drugs.

Compared with PDA-AC3, the major change in NC3 was more pores after carbonization. The specific surface area of NC3 increased to 433 m^2^/g, and its drug loading performance also improved. Interestingly, the adsorption capacity of NC3 reached 215.02 mg/g, which is close to that of AC3 (246.44 mg/g) although the specific surface area and pore volume of NC3 are only half those of AC3, suggesting that the chemical interaction may also contribute to the adsorption capacity.

### 3.4. The Zeta Potential Analysis

The zeta potential measures the residual charge on the surface of the AC and the loaded AC. The AC shows a negative zeta potential, and 2,4-D sodium exists as an anion in aqueous solution. When AC adsorbs 2,4-D sodium, it has a higher negative zeta potential, indicating that AC successfully adsorbs 2,4-D sodium (Figure 4).

After modification, the electronegativity of activated carbon PDA-AC3 and NC3 decreased obviously, indicating that positive groups were introduced. The modified amino-NH_2_ is non-charged and can combine with H^+^ ionized from water to form a positively charged state-NH_3_^+^.

### 3.5. The Release of 2,4-D Sodium

Figure 5 shows a comparison of the sustained release capacities of the five AC preparations for 2,4-D sodium at 25 °C. The five types of AC released 2,4-D sodium faster in the first 24 h. After 96 h, the dissolution equilibrium is essentially reached, after which the release is slower. The release rate of the AC1 preparation is only 32.31%, that of the AC2 preparation is 38.15%, and that of AC3 is 51.07%. However, the release capacity of modified PDA-AC3 is significantly improved, and the loaded drug could be released completely within 3 days. Through modification with dopamine hydrochloride, the micropore size decreases and the mesopore volume increases, which enhances the desorption ability of drugs and carriers. At the same time, chemical groups such as amines are introduced after modification, which makes the bound 2,4-D sodium easier to fall off, so the release rate of PDA-AC3 is improved. In addition, dopamine hydrochloride enhances the water solubility of activated carbon and makes it better dispersed in water solution. Meanwhile, the PDA enhanced surface charges properties when 2,4-D loaded on the PDA-AC3 and NC3, with more charge interaction between carriers and the drug, which is helpful to improve the release property.

As is shown in Figure 5, the slow-release performance of nitrogen-doped activated carbon NC3 was partly improved, and the release rate of NC3 was prolonged, when reaching the equilibrium of slow-release in 10 days, and the release rate is 48.6%. The above results suggest that physical interaction is the main factor affecting the drug adsorption capacity of activated carbon carriers, and pore size distribution and chemical interaction are the main factors affecting the drug release performance of activated carbon carriers. This is helpful to improve drug absorption and release capacity by adjusting pore size distribution or introducing chemical groups.

## 4. Conclusions

The five AC carriers show good adsorption and sustained release effects on 2,4-D sodium, indicating that AC is a good carrier for delivering pesticides. AC1, with the highest specific surface, possesses the best drug-loading capacity, reaching 679.18 mg/g, but the lowest drug release rate of 32.31% in 96 h. PDA-AC3 has the lowest specific surface, showing limited drug-loading ability, 82.94 mg/g, but complete drug release within 72 h. The drug-loading capacity of nitrogen-doped activated carbon NC3 recovered to 215.02 mg/g, which is similar to that of AC3 (246.44 mg/g), although the specific surface area and pore volume of NC3 are almost half those of AC3. These results suggest that the pore size distribution and chemical interaction are major factors affecting the drug loading and release of activated carbon carriers. The optimized design of appropriate pore size distribution and modification of chemical groups on AC carriers will help us to obtain new drug carriers with excellent adsorption and controllable release properties.

## Figures and Tables

**Figure 1 materials-12-04019-f001:**
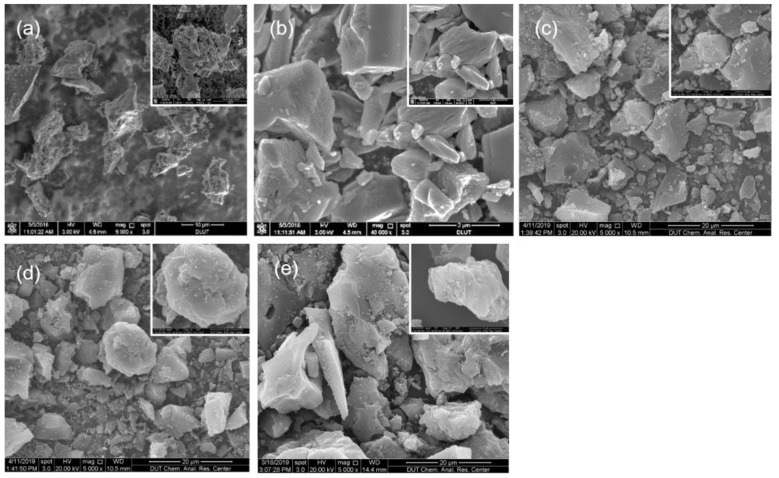
Scanning electron microscope image of the activated carbons: (**a**) AC1; (**b**) AC2; (**c**) AC3; (**d**) PDA-AC3; and (**e**) NC3.

**Figure 2 materials-12-04019-f002:**
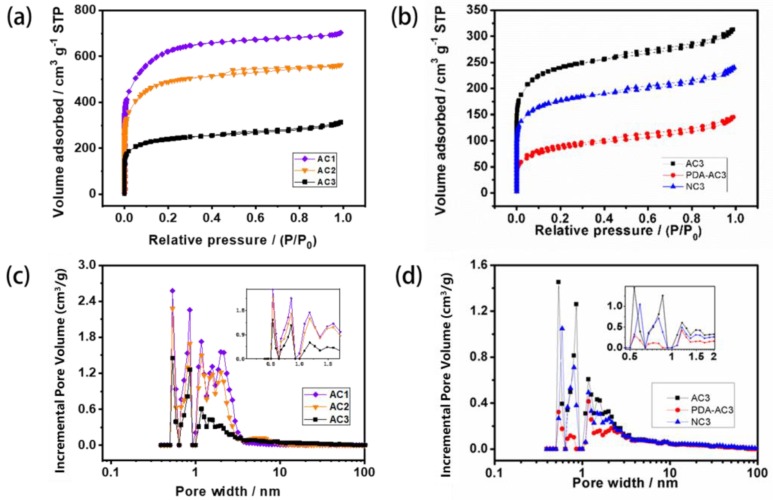
N2 adsorption−desorption isotherm of different activated carbons: (**a**) AC1, AC2, AC3; (**b**) AC3, PDA-AC3, NC3; Pore size distributions for different activated carbons calculated using DFT (density functional theory) software: (**c**) AC1, AC2, AC3; (**d**) AC3, PDA-AC3, NC3.

**Figure 3 materials-12-04019-f003:**
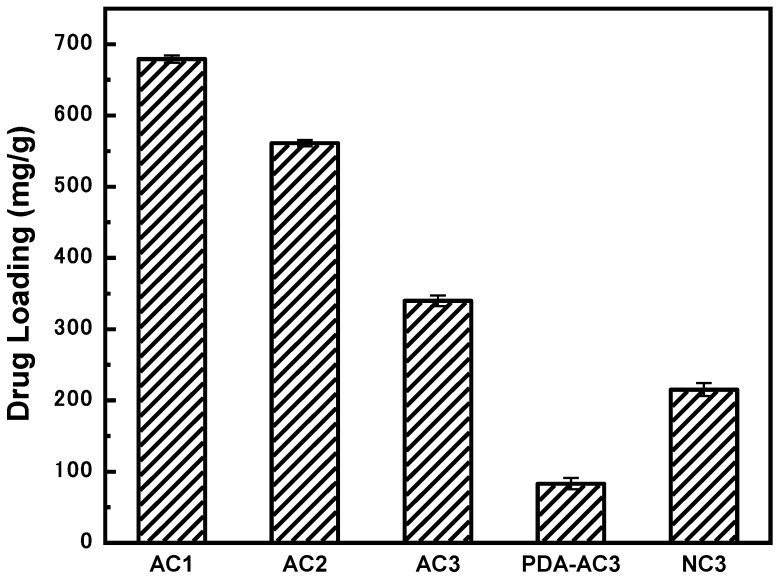
Adsorption capacity of different activated carbon materials for 2,4-D sodium.

**Figure 4 materials-12-04019-f004:**
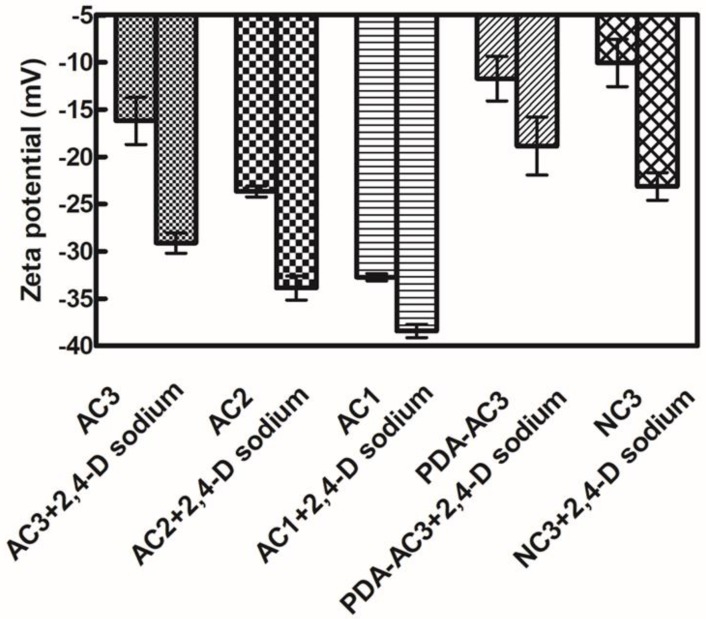
Zeta potential of different activated carbons with 2,4-D sodium.

**Figure 5 materials-12-04019-f005:**
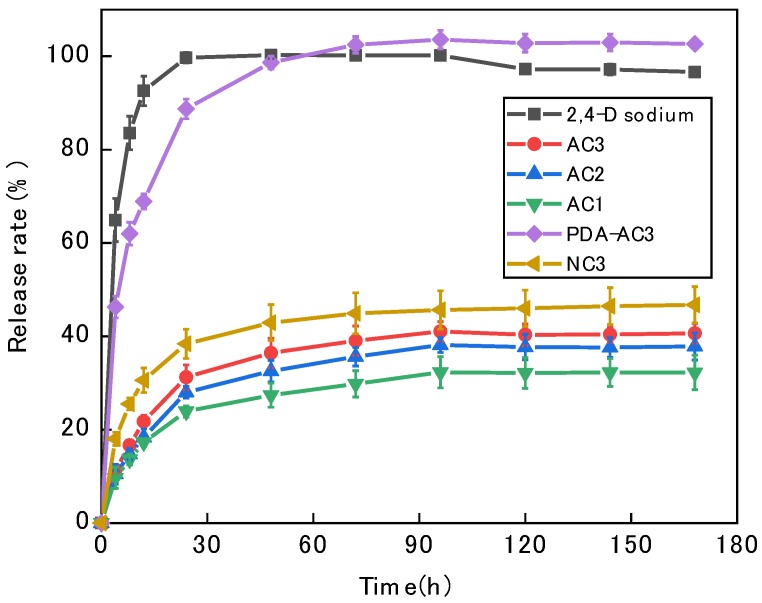
Release rate of 2,4-D sodium from different drug-loaded activated carbon materials.

**Table 1 materials-12-04019-t001:** Specifurface area and pore size parameters of different activated carbons.

Samples	S_BET_ (m^2^/g)	S_meso_ (m^2^/g)	S_mic_ (m^2^/g)	V_total_ (cm^3^/g)	V_mic_ (cm^3^/g)
AC1	2248	1079	1169	1.09	0.48
AC2	1801	685	1116	0.87	0.45
AC3	877	307	570	0.49	0.24
PDA-AC3	230	192	38	0.22	0.05
NC3	433	246	187	0.37	0.17

**Table 2 materials-12-04019-t002:** Elemental analysis before and after modification.

Samples	Carbon%	Nitrogen%	Hydrogen%
AC3	88.15	0.636	0.309
PDA-AC3	84	1.997	0.717
NC3	90.8	1.694	0.215

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
