# Peer review of "The Enhanced and Tunable Sustained Release of Pesticides Using Activated Carbon as a Carrier"

_materials, 2019, doi:10.3390/ma12234019_

Round 1
Reviewer 1 Report
Authors presented in paper the five AC carriers show good adsorption and sustained release effects on 2,4-D sodium and indicating that AC is a good carrier for delivering pesticides in agricultural pest control as an inexpensive, effective and safe pesticide carrier. The test material and its characteristics are made correctly. The description lacks justification and discussion on the role of dopamine in the carbon material structure formation process. The process of pesticide release under laboratory conditions is significantly different from processes in soil. Transferring this data beetwen laboratory and enviromental is unreliable, this process can be controlled by soil or water pH. The publication does not discuss the possibility of converting the 2.4D salt into acid (role of pH). The results of the work are not innovative, and the results obtained do not bring significant scientific news.
Reviewer 2 Report
This manuscript addesses the practical use with activated carbon.
Authors demonstrated the use of ACs as a carbon-based vector for a model pesticide. Overall, the work is pedestrian and would be only interested in a very narrow audience who want to modify AC with dopamine. I would like authors to substantially revise the manuscript including discussion on more scientific points and investigation on structures and physical properties.
Several technical comments are as follows.
1. They used PDA to give intermolecular interaction with a model pesticide, but it turned out to fairly decreased loading capacity. The explanation should be given.
2. What is the actual structure of NC3? How does it change after thermal treatment from PDA-AC3?
3. Please detail the control sample in Figure 5?
4. In table 1, It would be interesting if authors discuss the effect of mesoporosity given that PDA-AC3 showed the high amount of mesopores.
5. why did the percentage of nitrogen decrease in NC3?
Reviewer 3 Report
Peer review report
The manuscript reports on usage of modified activated carbon as a carrier for the release of herbicide 2, 4-D sodium. The reported work could be of interest to researchers working in the field of materials and environmental remediation. The authors need to address the following before the paper could be recommended for publication.
The introduction part needs to be modified to include more information on the problems created by the herbicide. What are the particular health effects? It says “However, the drug release capacity of AC has been 42 less well studied, which limits its application as a drug or pesticide carrier. I think it would be wrong to say this and authors should cite previous literature on this to make a strong case for their current investigations.
More detailed information in the characterization section (section 2.5) is required for a better understanding.
Consider checking the language throughout the manuscript. There are several mistakes.
Check the y-axis label of figures 2 b.
It says “The pore sizes of the five activated 116 carbons are mainly distributed in the regions of 0.5-2 nm”. However, the average pore diameters presented in Table 1 are different.
Please enhance the quality of the sections abstract and conclusions by including the numbers from the adsorption and release of pesticide.
Round 2
Reviewer 1 Report
The corrections and additions made by the authors mean that the publication can be accepted. The scientific problem discussed in the publication is important for a wide range of chemists, especially those dealing with environmental protection.